# High front-to-back ratio metasurface antenna with single-layer and compact size for WLAN applications

**Tan Dao-Duc, Noi Truong-Quang, Tu Le-Tuan, Hung Tran-Huy**[ORCID]*

Faculty of Electrical and Electronic Engineering, PHENIKAA University, Hanoi, Vietnam

* hung.tranhuy@phenikaa-uni.edu.vn

**Data availability statement:** All relevant data are within the manuscript.

## Abstract

This paper proposes a novel high front-to-back ratio (FBR) metasurface (MS) antenna with single-layer and compact size characteristics. The proposed design consists of a $2 \times 2$ unit-cell MS radiating layer, which is capacitively coupled with a directed-fed crossed patch in the same layer. Accordingly, compact design with single-layer MS antenna can be obtained. It is worth noting that the proposed configuration is different from the conventional MS antenna, which is commonly designed with multiple layers, leading to high profile and complicated structure. Finally, a substrate integrated waveguide cavity is employed to improve the FBR of the proposed design. A prototype with overall dimensions of $0.52 \lambda \times 0.52 \lambda \times 0.03 \lambda$ achieved measured operating bandwidth of about 10.6%. Despite having compact size, the proposed antenna always performs high broadside gain of better than 5.2 dBi and high front-to-back ratio of better than 16 dBi entire the operating bandwidth. These operation characteristic distinguishes the proposed design from the other MS antennas with more complicated structures.

## Introduction

To come up with modern wireless communications, simple and compact antennas with high performance are demanded. Among various types of antenna structures including dipole/monopole, slot, horn, Vivaldi, and so on, microstrip patch antenna is one of the best solutions for low cost, lightweight, as well as simple structure. However, traditional patch antenna suffers from a relatively narrow operating bandwidth of less than 3%.

Conventional method to increase the bandwidth of microstrip patch antenna is to introduce additional resonances, which are in proximity to each other. Wideband performance can be achieved by using multi-layer structures [1,2] or L-probe proximity-fed scheme [3]. Overall, the main drawbacks of such antennas include multilayer and high-profile structure. In [4–9], lower profile with single-layer designs are reported, in which a capacitively feeding technique is utilized or multiple slots are inserted into the main radiating patch. However, limited operating bandwidths are the drawback of these designs. Alternatively, positioning multiple parasitic elements around the radiating patch is also an effective method for bandwidth enhancement [10–12]. Although wideband performance can be achieved, the use of parasitic elements significantly increases the antenna lateral dimensions.

**Funding:** The author(s) received no specific funding for this work.

**Competing interests:** The authors have declared that no competing interests exist.

In recent years, artificial magnetic conductor (AMC) or metasurface (MS) structures have been developed to achieve wideband operation. In [13–15], the AMC is used as a substitute for metal ground to achieve both antenna miniaturization and bandwidth enhancement. On the other hand, the MS is generally capacitively coupled through slots to excite different operating modes for wideband performance [16–20]. By loading additional capacitance, compact structure can be realized. However, these kinds of antennas with AMC or MS have the disadvantages of high profile and multi-layer configurations. Besides, MS antennas also exhibit low FBR due to the unwanted radiation from the slots in downwards direction.

In this paper, a novel MS antenna with single-layer and compact size while achieving high FBR is presented. The antenna has an MS layer with $2 \times 2$-unit cells acting as the primary radiating element. Instead of exciting the MS layer through slots, the proposed MS layer is excited by a crossed patch in the same layer. This kind of configuration allows the proposed antenna to be designed with a single layer. Finally, the antenna is surrounded by a substrate integrated waveguide (SIW) cavity for front-to-back ratio improvements, especially in the high frequency range.

## Narrow-band MS antenna

### MS layer design

For MS antenna, MS layer plays an important role in determining the radiation characteristics of the whole antenna. The operating resonance strongly depends on the dimensions of the cell (W) and the periodicity (P). There are different approaches to investigate the operation of the MS layer, such as dispersion diagram, characteristic mode analysis (CMA). On the one hand, the dispersion diagram investigates the characteristics of single unit cell. On the other hand, CMA can exactly predict the dominant mode on the whole MS structure [21,22].

CMA is carried out for an array of $2 \times 2$ MS units at 5.0 GHz using the MoM-based CMA tool in the commercial simulation software CST MWS. The modal significance of the first four modes is presented in Fig 1. According to the CMA theory, the mode resonates and radiates the most efficiently when its modal significance is equal to 1. As observed, in the frequency range around 5.0 GHz, the modal significances of Mode 1 and Mode 2 are identical, which are approximately 1. It is worth noting that the other modes with low modal significances radiate inefficiently at this frequency range. Besides, the surface current and 3-D radiation pattern of Mode 1 and Mode 2 depicted in Fig 2 also demonstrate that the MS structure can provide good broadside direction around 5.0 GHz. For Mode 3 and 4, the currents flowing on the MS are in different directions, leading to the omni-directional beam. Accordingly, they are not suitable for the design target of uni-directional beam antenna.

### Antenna design and performance

Fig 3 shows the geometrical configuration of the proposed single-layer MS antenna. The antenna is printed on the top side of the Taconic RF-35 substrate (dielectric constant of 3.5 and loss tangent of 0.003) and directed fed through a 50-$\Omega$ SMA. The MS layer consists of an array of $2 \times 2$ units. To excite the MS, a rectangular patch is located at the center of the MS layer, and the feeding position will be located on this patch. The optimized design parameters are as follows: $W_s = 27$, $P = 13.5$, $W = 12.5$, $g = 0.5$, $l_p = 14.6$, $w_p = 2$, $l_f = 2.6$ (unit: mm).

The simulated performance in terms of reflection coefficient and realized gain of the proposed single-layer MS antenna is shown in Fig 4. The simulated matching bandwidth with reflection coefficient of less than –10 dB is from 5.0 to 5.3 GHz, corresponding to about 5.8%. Regarding the broadside gain, the gain across the operating spectrum is from 6.5 to 7.0 dBi.

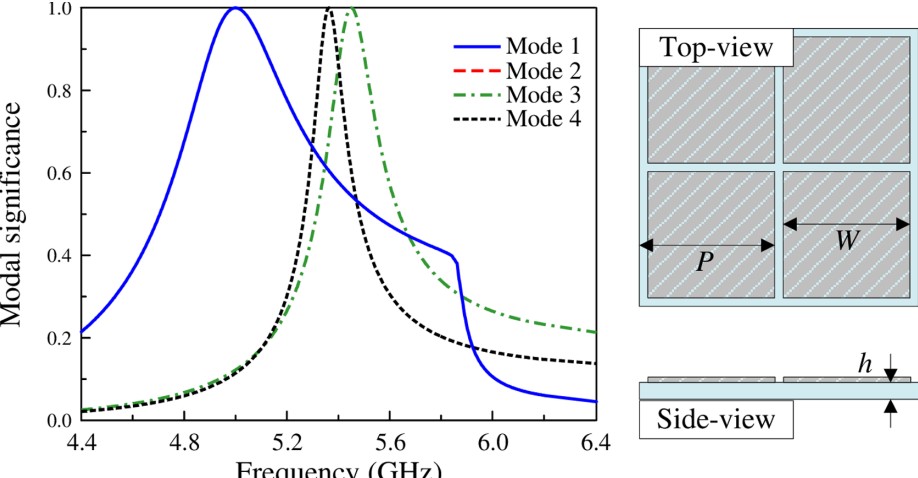

**Fig 1. Simulated modal significance of the array of 2 × 2 MS units.**

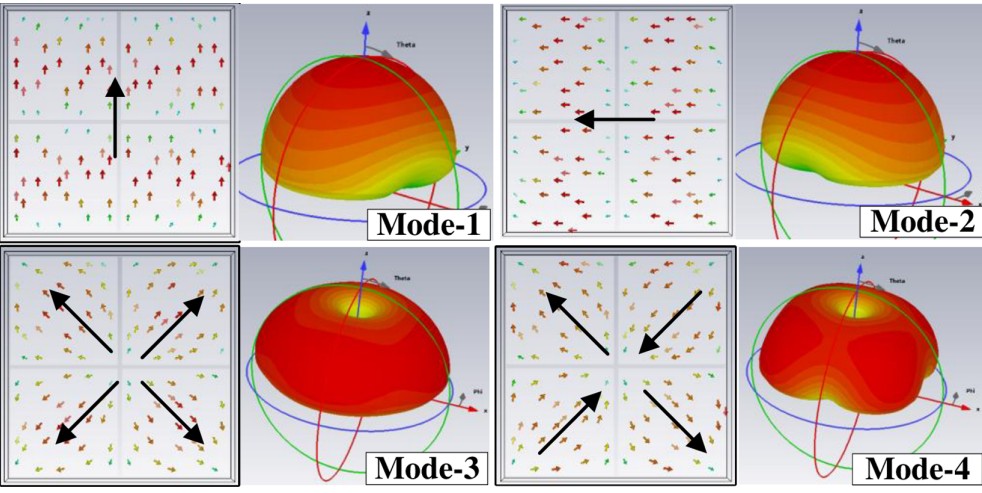

**Fig 2. Simulated surface currents and 3-D radiation patterns of Mode 1 and Mode 2.**

These simulated data is quite consistent with the CMA on this antenna, shown in Fig 5. Noted that only the modes with broadside radiation patterns in the investigated frequency band are considered. Besides, the CMA is implemented without coaxial cable excitation. As seen, Mode 1 and 2 have different resonances due to the asymmetrical geometry. Here, Mode 1 is utilized and when the excitation feed is positioned on the center rectangular patch, the antenna can achieve good operation, as demonstrated in Fig 4.

## Antenna operation characteristics

To understand the operating principle. The equivalent circuits of the conventional MS layer and the presented MS layer are depicted in Fig 6. The conventional MS can be modelled as a parallel $LC$ circuit, in which $L_d$ is the ground dielectric slab inductance and $C_o$ is the edge capacitance. Accordingly, the resonant frequency ($f_r$) can be calculated based on the following

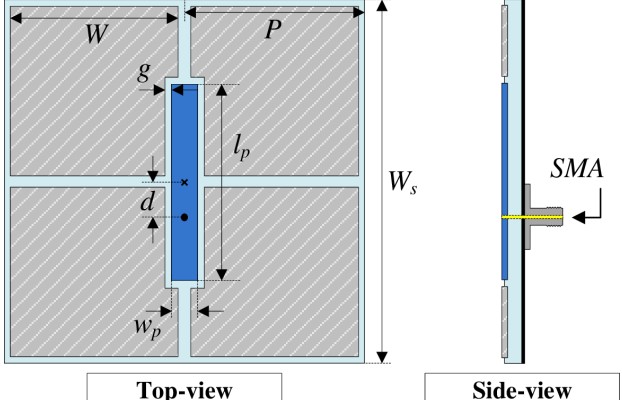

**Fig 3. Geometrical configuration of the single-layer MS antenna.**

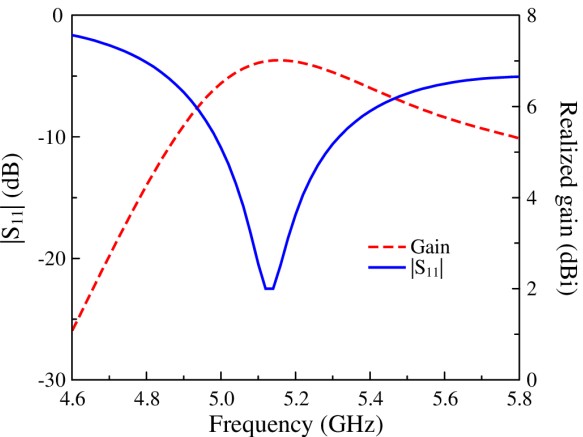

**Fig 4. Simulated reflection coefficient and realized gain of the MS antenna.**

equation:

$$f_r = \frac{1}{2\pi\sqrt{L_d C_o}} \tag{1}$$

The MS with slotted feed is equivalent to a parallel $L_d C_o$ circuit. Meanwhile, the proposed MS antenna provides more capacitances, which can help to adjust the operating frequency conveniently. For demonstration, Fig 7 presents the simulated $|S_{11}|$ of the proposed single-layer MS antenna for different values of $g$ and $W$. As observed, increasing $g$ results in smaller capacitance of $C_2$ and $C_3$, leading to smaller total capacitance as well. According to Equation (1), the resonant frequency will be increased. This is confirmed by observing the simulated data when varying $g$. With respect to the unit cell size, which determines the operating resonance, increasing $W$ leads to lower operating frequency range. As seen in the remaining plot in Fig 7, when $W$ varies from 12 to 13 mm, the operating frequency shifts from 5.3 to 4.9 GHz. Further investigation on the effectiveness of the excited patch on the reflection coefficient is shown in Fig 8. The data indicates that changing $l_p$ has minor effect on the resonant frequency. Varying $l_p$ from 13.4 to 15.2 mm, the resonance is just around 5.1 GHz. To conclude, it can be demonstrated that the size of the unit cell ($W$) and the gap between

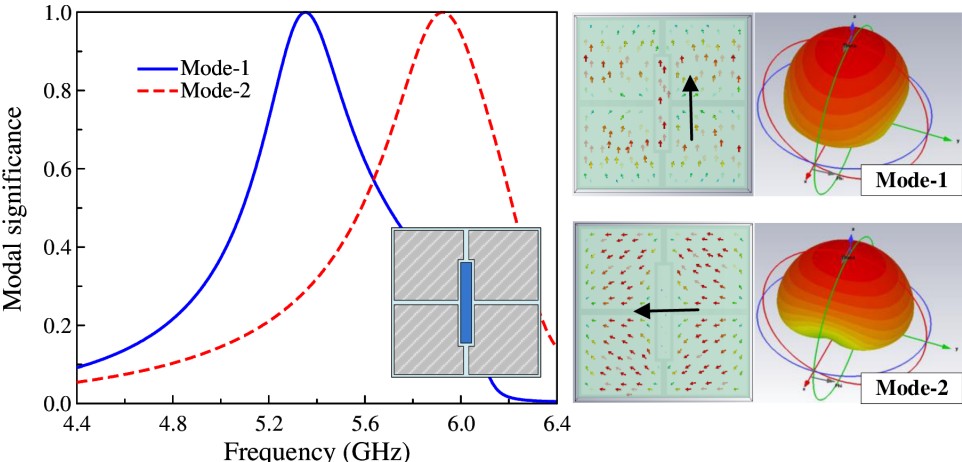

**Fig 5. CMA results of the proposed single-layer MS antenna.**

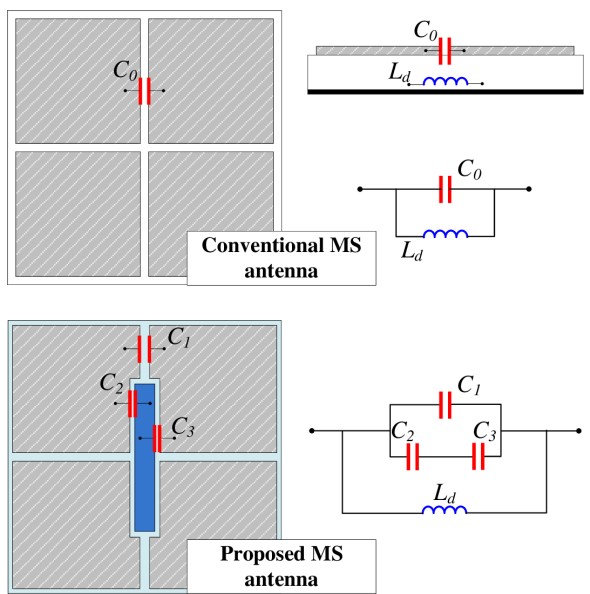

**Fig 6. Equivalent circuits of different MS antennas.**

the excited patch and the unit cell ($g$) are critical parameters to the operating band of the proposed single layer MS antenna rather than the length of the excited patch ($l_p$).

## Wideband MS antenna

As discussed in the previous section, the operating bandwidth of the single-layer MS antenna is about 5.8%, ranging from 5.0 to 5.3 GHz. To expand the bandwidth, the excited source is modified from the rectangular patch to crossed patch by adding additional horizontal rectangular patch. Fig 9 shows the geometrical configuration of the proposed single-layer wideband MS antenna. The optimized design parameters are as follows: $W_s = 27$, $P = 13.5$, $W = 13.3$, $g = 0.4$, $l_{p1} = 16.6$, $l_{p2} = 17.6$, $w_p = 2.8$, $l_f = 3.7$ (unit: mm).

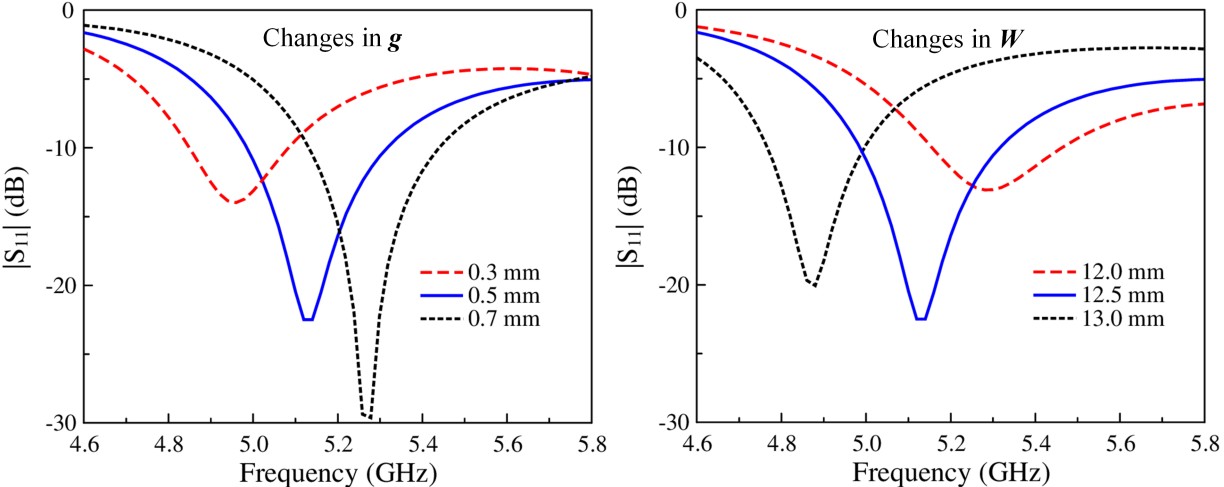

**Fig 7. Simulated $|S_{11}|$ for different values of $g$ and $W$.**

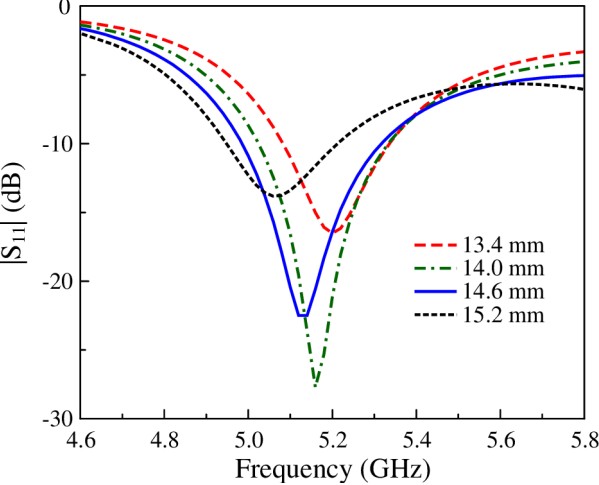

**Fig 8. Simulated $|S_{11}|$ for different values of $l_p$.**

To broaden the operating bandwidth, the principle is to produce additional resonance, which is proximity to the original resonance. Here, the crossed patch is used to excite another resonance of the MS layer. Fig 10 shows the simulated performance of the proposed wideband MS antenna. It is obvious that wideband performance can be achieved with two resonances in the $|S_{11}|$ profile. Compared to the MS antenna with rectangular excited patch, the –10 dB impedance bandwidth increases from 5.8% (5.0–5.3 GHz) to 10.6% (4.9–5.45 GHz). For better understanding the wideband operation mechanism, a CMA on this antenna type is shown in Fig 11. Here, four different modes with broadside beam are observed. Mode 1 and 2 have current flowing along x- and y- direction. Meanwhile, the currents of Mode-3 and 4 are diagonal.

For verification, Fig 12 shows the simulated current distribution on the MS layers at two resonances in the $|S_{11}|$ profile, 5.1 and 5.3 GHz. As observed, the current flows in the vertical direction at 5.1 GHz. Meanwhile, the current flows in the diagonal direction at 5.3 GHz

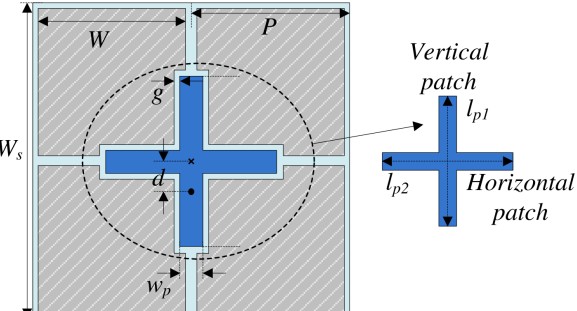

**Fig 9. Geometry of the proposed single-layer wideband MS antenna.**

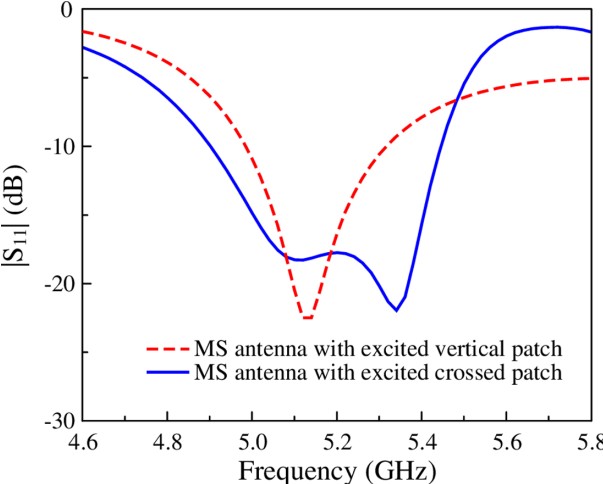

**Fig 10. Simulated $|S_{11}|$ of the proposed single-layer wideband MS antenna.**

with the presence of the horizontal patch. These current distributions are consistent with the CMA on this antenna. Further demonstration about the effect of the horizontal patch on the impedance bandwidth of the proposed antenna is shown in Fig 13. It can be seen clearly that the higher resonance is strongly affected by the horizontal patch.

## Wideband MS antenna with SIW cavity

Theoretically, the compact antenna often suffers from a critical drawback of high back radiation, which significantly degrades the FBR. This is due to the high diffracted power at the edge of the substrate. For the proposed structure, a SIW cavity is employed to prevent the diffraction of the surface wave at the edges of the ground plane. Accordingly, high FBR can be achieved. The SIW cavity requires multiple vias in proximity to form an electric wall, which is better with a greater number of vias. Meanwhile, the distance between the vias depends on the radius of the vias. The geometry of the final antenna design is shown in Fig 14. The optimal dimensions are as follows: $W_s = 29.6$, $P = 13.3$, $W = 13$, $g = 0.4$, $l_{p1} = 16.8$, $l_{p2} = 17.2$, $w_p = 2.6$, $l_f = 4.4$, $s = 1$, $w_a = 0.5$, $r_v = 0.1$, $d_v = 1.16$ (unit: mm).

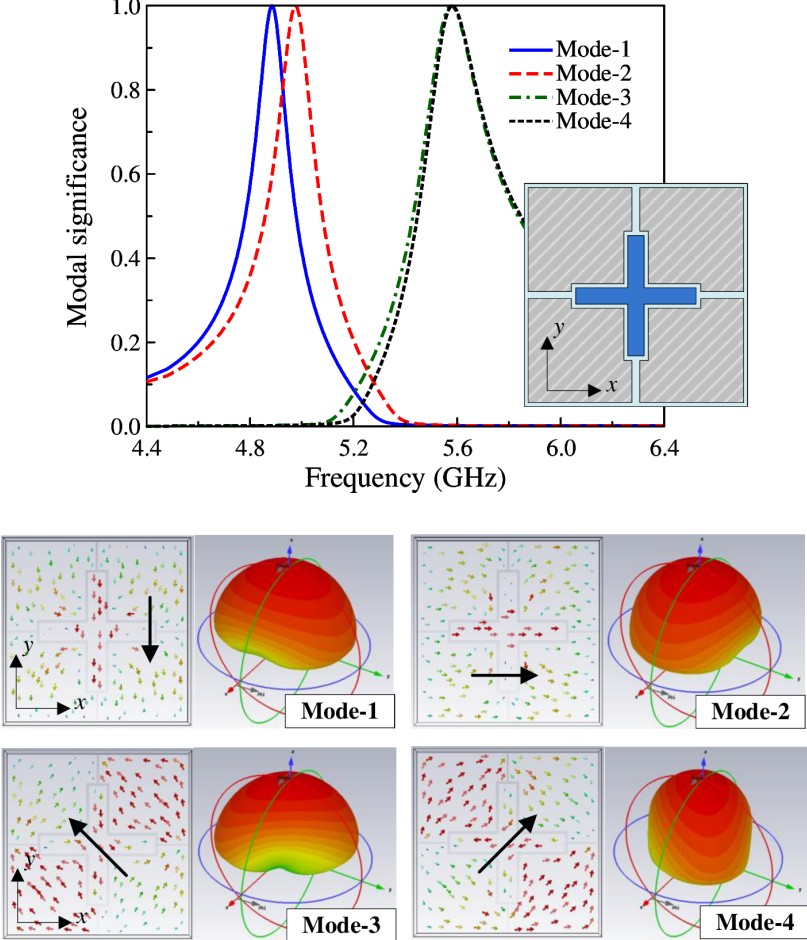

**Fig 11. CMA results of the proposed wideband single-layer MS antenna.**

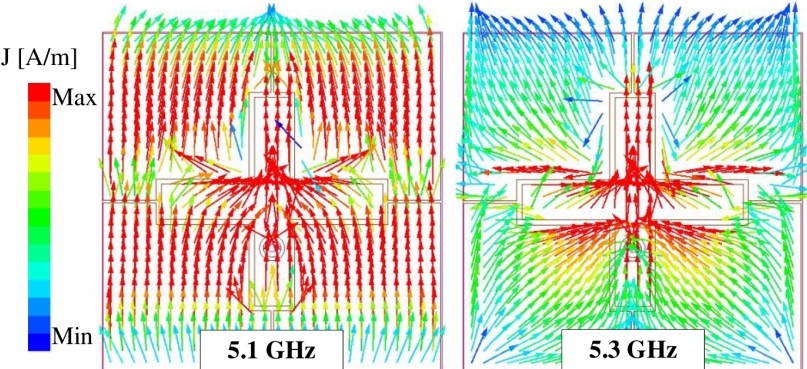

**Fig 12. Simulated current distributions of the proposed single-layer wideband MS antenna.**

To demonstrate the advantage of the proposed antenna in FBR enhancement, three different structures with similar overall dimensions are compared. The first one is the conventional microstrip patch antenna. The second antenna is the structure discussed in the previous section. The final one is the antenna with additional SIW cavity. Performance comparison in terms of $|S_{11}|$ and realized gain in both forward and backward directions is shown

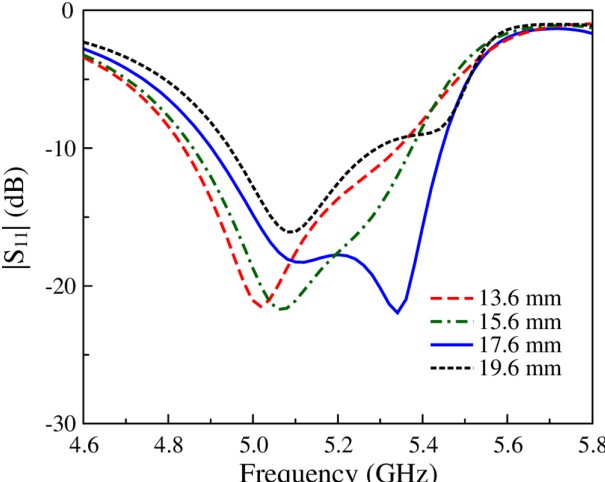

**Fig 13. Simulated $|S_{11}|$ of the proposed single-layer wideband MS antenna for different lengths of the horizontal patch, $l_{p2}$.**

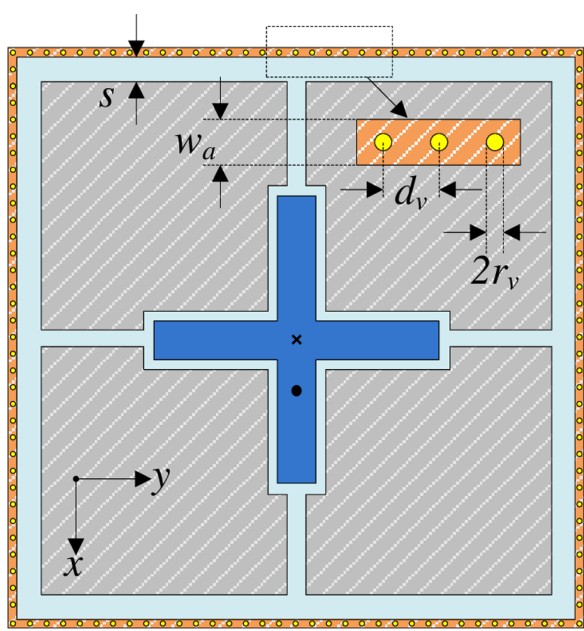

**Fig 14. Geometry of the proposed single-layer wideband MS antenna with SIW cavity.**

in Fig 15. As observed, the conventional patch antenna exhibits narrow impedance bandwidth with only one resonance in the $|S_{11}|$ profile. Meanwhile, the proposed MS antennas perform wider bandwidth with two adjacent resonances in the $|S_{11}|$ profiles. For the realized gain response, the conventional patch antenna has high back radiation, which is about –7 dB around 5.0 GHz. For the MS antenna with excited crossed patch, the back radiation is further suppressed in the frequency range of lower than 5.25 GHz. In higher range, the back radiation increases up to –7 dB at 5.4 GHz. With the presence of the SIW cavity, the back radiation can be adjusted and within the operating band from 5 to 5.52 GHz, the backwards power is in

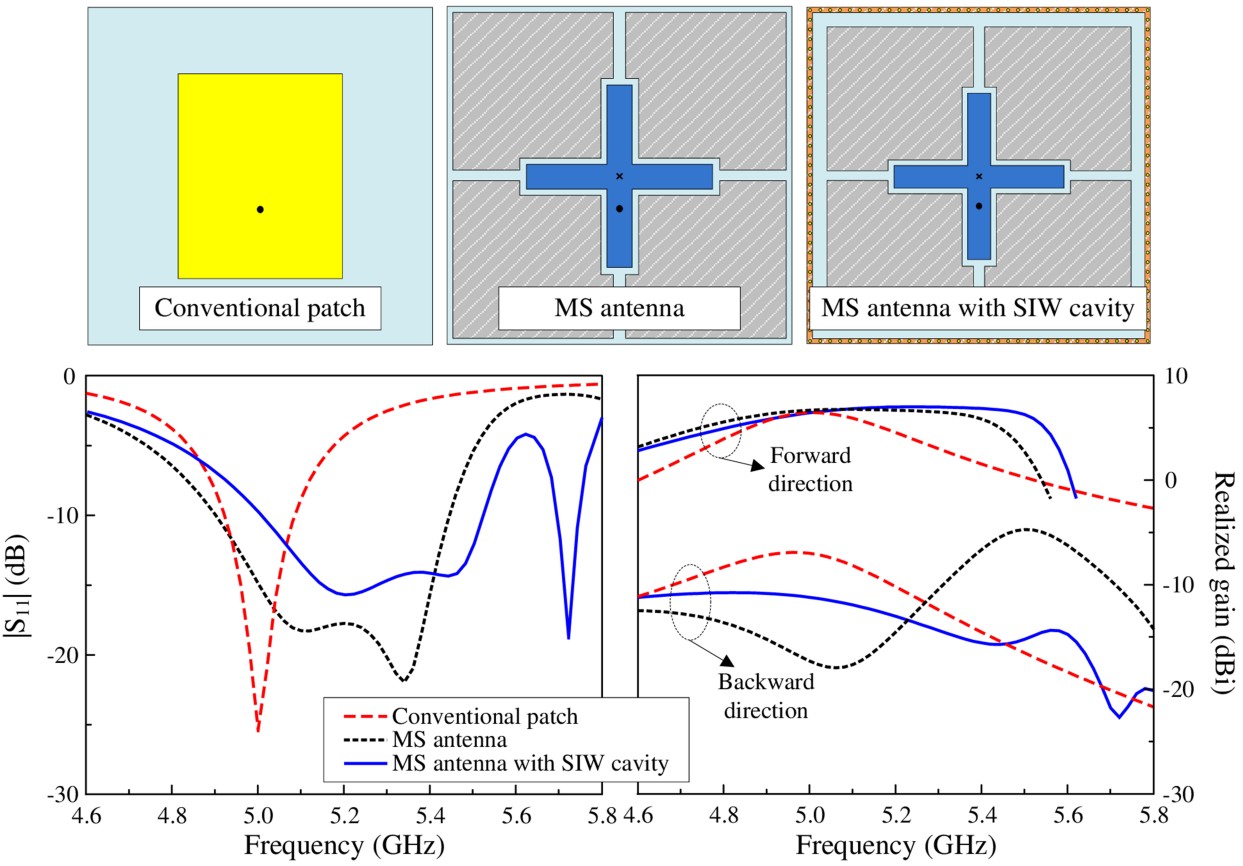

**Fig 15. Different antenna configurations and simulated performance of the proposed antenna regarding $|S_{11}|$ and realized gain.**

the range from –11.2 to –15.7 dB. According to these results, the function of the SIW can be demonstrated as an effective scheme to adjust the back radiation of the proposed MS antenna.

The simulated electric fields (E-field) on the plane containing the ground of such antennas are depicted in Fig 16. For the conventional patch antenna, the E-field at 5.0 GHz is quite strong at the edges of the substrate, leading to high back radiation. For the MS antennas at 5.4 GHz, the antenna without SIW cavity suffers from strong E-field at the boundary of the substrate. In contrast, the E-field is mostly occupied within the SIW cavity, resulting in small back radiation. Here, the power is mostly captured by the cavity, which is illustrated by the strong field distribution on the shorting pins of the cavity.

## Experimental results and discussion

An antenna prototype of the proposed single-layer wideband MS antenna with SIW cavity is fabricated and measured to validate the design concept and simulated results. Fig 17 shows the photographs of the fabricated antenna prototype.

The simulated and measured reflection coefficients are presented in Fig 18. It can be seen clearly that the measured impedance bandwidth for reflection coefficient lower than –10 dB is 10.6% (4.99–5.55 GHz). Across this band, the measured broadside gain depicted in Fig 19 is always better than 6.0 dBi. The data in Fig 19 also indicates that the proposed antenna

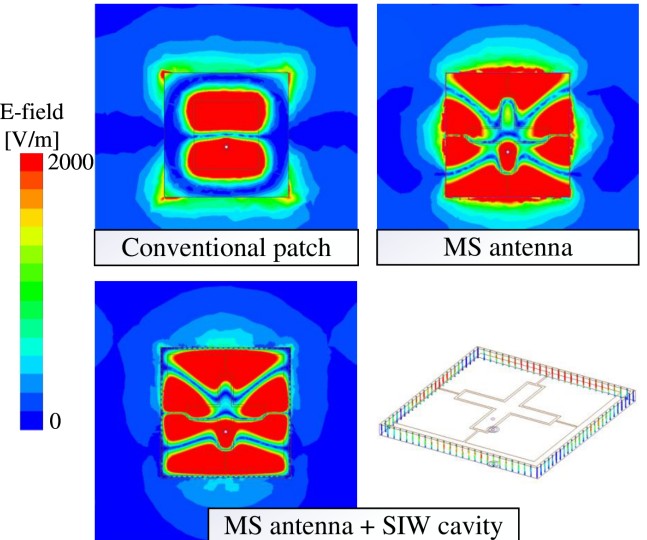

**Fig 16. Simulated E-field distributions of different antennas.**

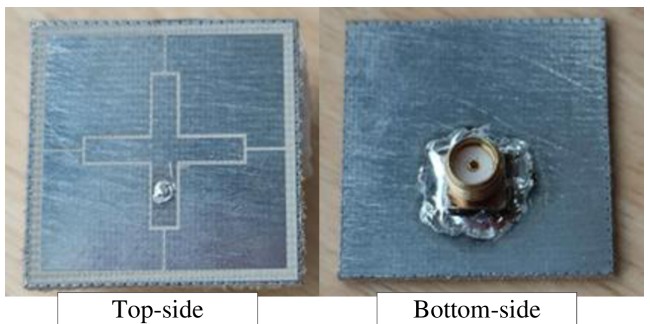

**Fig 17. Photographs of the fabricated antenna.**

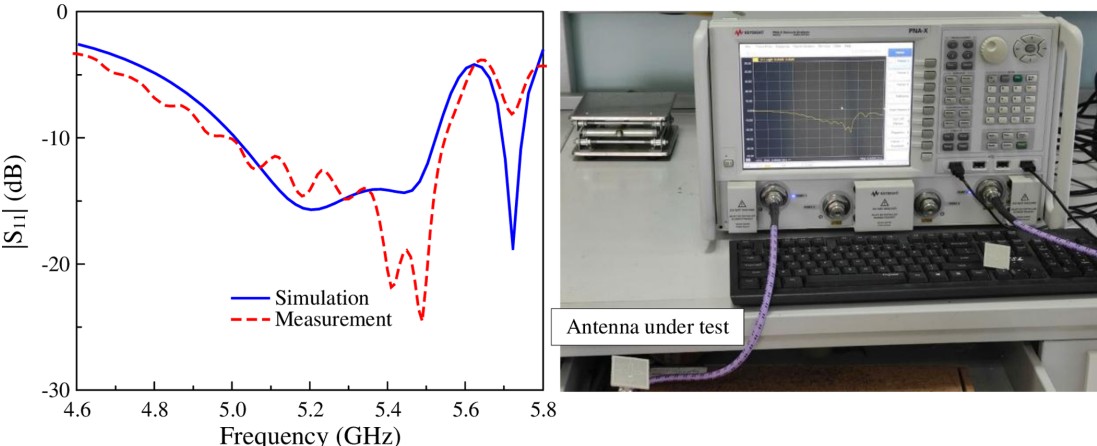

**Fig 18. Simulated and measured $|S_{11}|$ of the proposed antenna.**

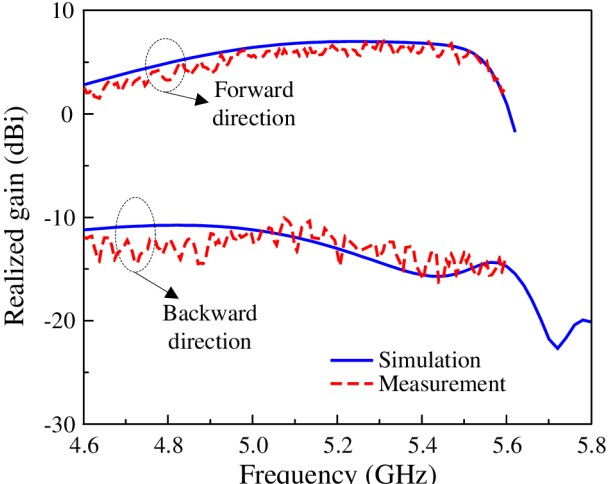

**Fig 19. Simulated and measured realized gain of the proposed antenna.**

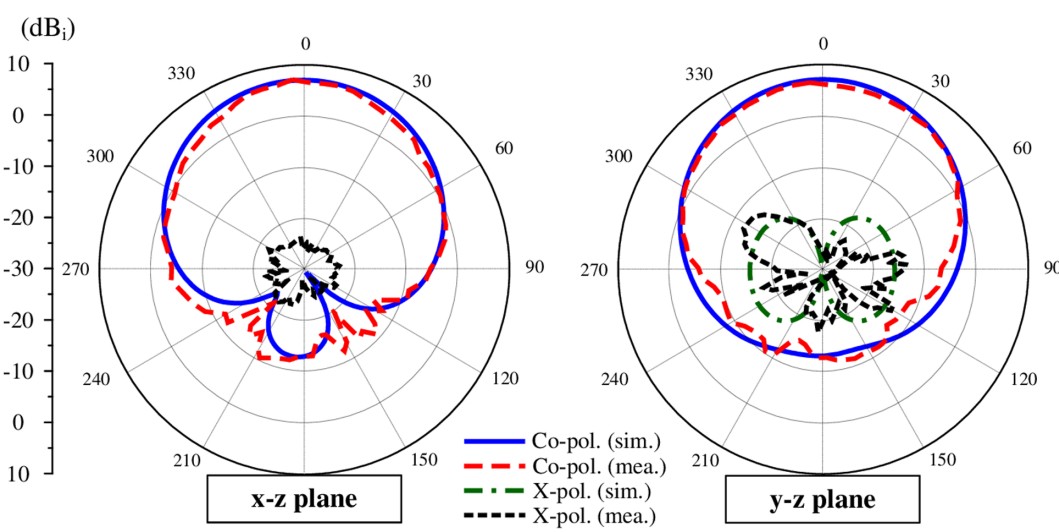

**Fig 20. Simulated and measured gain radiation patterns of the proposed antenna at 5.2 GHz.**

can achieve high FBR of greater than 16 dB. Fig 20 plots the simulated and measured co-polarization (Co-Pol) and cross-polarization (X-Pol) at 5.2 GHz in *x–z* and *y–z* planes. As observed, the radiation patterns are symmetric around the boresight direction and stable across the impedance bandwidth. Besides, the X-Pol level is significantly lower than the Co-Pol level, which is about 40 dB difference. In terms of radiation efficiency, the simulated result indicate that the proposed antenna can achieve the radiation efficiency of better than 85% within the operating bandwidth.

The advantages of the proposed antenna can be demonstrated by making comparison with the related works, as exhibited in Table 1. It can be seen obviously that high FBR, simple structure with single-layer design, low-profile and compact configuration are the main advantages of the proposed antenna. Compared with the single-layer design, it can be seen that using parasitic elements can improve the bandwidth, but the designs in [11,12] have very

**Table 1. Comparison with related designs.**

| Ref. | Overall size ($\lambda$) | No. of layers | Structure | Bandwidth (%) | Max. Gain (dBi) | Min. FBR (dB) |
|------|--------------------------|---------------|-----------|---------------|-----------------|---------------|
| [5] | $0.82 \times 0.66 \times 0.03$ | 1 | Capacitive fed patch | 8.4 | 7.3 | 15 |
| [7] | $0.86 \times 0.86 \times 0.03$ | 1 | Patch with slots | 6.8 | 7 | 15 |
| [11] | $1.20 \times 1.20 \times 0.03$ | 1 | Patch + parasitic elements | 20.1 | 9.5 | 16 |
| [12] | $1.61 \times 1.00 \times 0.08$ | 1 | Patch + parasitic elements | 41.8 | 10.5 | 15 |
| [15] | $1.01 \times 0.75 \times 0.05$ | 3 | Patch + AMC ground | 32.9 | 10.2 | 18 |
| [18] | $0.73 \times 0.71 \times 0.06$ | 3 | MS + slotted feed | 24.5 | 7.8 | 11 |
| [19] | $0.41 \times 0.41 \times 0.09$ | 3 | MS + slotted feed | 44.4 | 6.5 | 10 |
| [20] | $0.42 \times 0.42 \times 0.07$ | 3 | MS + slotted feed | 43.1 | 6.1 | 6 |
| Prop. | $0.52 \times 0.52 \times 0.03$ | 1 | MS + crossed patch feed | 10.6 | 6.4 | 16 |

large dimensions. Compared with the similar MS designs [15,18–20], the presented work has the smallest number of layers, lowest profile, and the best FBR while maintaining compact structure. This is due to the different feeding scheme, which distinguishes the proposed work from the conventional MS antennas.

## Conclusion

The single-layer compact MS antenna with wideband and high FBR characteristics has been presented and investigated in this paper. The evolution process to achieve the final realization of the proposed antenna is also discussed thoroughly. The final design with compact dimensions of $0.52\,\lambda \times 0.52\,\lambda \times 0.03\,\lambda$ exhibits operating bandwidth of about 10.6%. Across this band, the broadside gain is always better than 5.2 dBi and the FBR is higher than 16 dBi. This design concept provides a feasible solution to realize a compact MS antenna with simple structure and good operating performance, which is potentially utilized in modern wireless systems.

## Author contributions

**Conceptualization:** Hung Tran-Huy.

**Data curation:** Noi Truong-Quang.

**Investigation:** Tan Dao-Duc.

**Methodology:** Tu Le-Tuan.

**Project administration:** Hung Tran-Huy.

**Supervision:** Hung Tran-Huy.

**Writing – original draft:** Tan Dao-Duc, Noi Truong-Quang.

**Writing – review & editing:** Tu Le-Tuan.

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
