## [Decision Letter · Decision Letter 0]

4 Feb 2025

PONE-D-25-03828High front-to-back ratio metasurface antenna with single-layer and compact size for WLAN applicationsPLOS ONE

Dear Dr. Tran-Huy,

Thank you for submitting your manuscript to PLOS ONE. After careful consideration, we feel that it has merit but does not fully meet PLOS ONE’s publication criteria as it currently stands. Therefore, we invite you to submit a revised version of the manuscript that addresses the points raised during the review process.

We look forward to receiving your revised manuscript.

Kind regards,

Sachin Kumar, Ph.D.

Academic Editor

PLOS ONE

Additional Editor Comments:

The reviewers agree that the paper is scientifically valid and interesting. However, there are several issues that must be addressed in order to improve the overall quality of the manuscript. Please read the reviewers' comments and respond to them properly.

Reviewers' comments:

Reviewer's Responses to Questions

**Comments to the Author**

1. Is the manuscript technically sound, and do the data support the conclusions?

Reviewer #1: Partly

Reviewer #2: Yes

2. Has the statistical analysis been performed appropriately and rigorously? 

Reviewer #1: N/A

Reviewer #2: Yes

3. Have the authors made all data underlying the findings in their manuscript fully available?

Reviewer #1: Yes

Reviewer #2: Yes

4. Is the manuscript presented in an intelligible fashion and written in standard English?

Reviewer #1: Yes

Reviewer #2: Yes

5. Review Comments to the Author

Reviewer #1: Authors need to work more in elaborating the working of antenna using CMA. Lot of literature work is available on CMA based antennas, how your work is different compared to previous recent work on CMA based metasurface antennas?

1. Please show MS surface current distribution of mode 3&4.

2. Why the surface current distribution in Fig.2 for MS1&2 is orthogonal?

3. How authors are able to determine the feeding point of antenna in Fig. 3.

4. Authors must show the MS graphs for different modes in case of Fig. 3 and Fig.8 antennas and please comment on them.

Reviewer #2: 1. The authors must describe the significant resonant modes for the proposed design.

2. Some parametric investigations on design parameters must be shown and discussed.

3. How the coupling between metacells exists, an explanation is desired.

4. What is the utility for asserting SIW technique?

5. A comparative study analysis with other literatures must be shown for radiation efficiency and number of antenna layers.

6. PLOS authors have the option to publish the peer review history of their article (what does this mean?). If published, this will include your full peer review and any attached files.

Reviewer #1: No

Reviewer #2: **Yes: **Asutosh Mohanty

---

## [Author Response · Author response to Decision Letter 1]

11 Feb 2025

Original Manuscript ID: PONE-D-25-03828

Original Article Title: “High front-to-back ratio metasurface antenna with single-layer and compact size for WLAN applications”

To: Reviewer

Re: Response to reviewer

Dear Reviewer,

We appreciate you for your precious time in reviewing our paper and providing valuable comments. It was your valuable and insightful comments that led to possible improvements in the current version. The authors have carefully considered the comments and tried our best to address every one of them.

We are uploading our point-by-point response to the comments, an updated manuscript with red highlighting indicating changes, and a manuscript without track changes.

Best regards,

Reviewer 1:

Concern # 1: Authors need to work more in elaborating the working of antenna using CMA. Lot of literature work is available on CMA based antennas, how your work is different compared to previous recent work on CMA based metasurface antennas?

Author response: It is noted that CMA is a tool to predict the resonance behavior of a whole metallic object. It means that if we have a metallic object with an arbitrary shape, using CMA can predict what are the dominant modes generated by this object. In this paper, the CMA is utilized to determine the possible operating resonances of the utilized 2 × 2 MS layer. Apart from this tool, using dispersion diagram analysis on a single unit-cell is another effective way to predict the resonances of the MS layer.

The authors agree with the Reviewer that lots of MS antennas have been reported in the literature. These MS-based antennas use CMA as a method to determine the operating frequency band of the utilized MS. The MS layers act as the parasitic elements, which are generally coupled with the primary radiating source (patch or slot).

Regarding the difference between the proposed MS and the other MS antennas, it has been discussed in Section “Introduction” that the reported MS antennas [16– 20] commonly designed with the MS and the excited source in different layers. Accordingly, multi-layer and bulky structures are the current drawbacks. Besides, high back radiation due to the use of slot as the primary radiating source is also another disadvantage. In this paper, authors proposed a single-layer MS antenna, which has an excited source and a MS in the same layer. This configuration allows the proposed antenna to be designed with a single layer. Besides, using SIW cavity also helps increase the front-to-back ratio. These features distinguish our proposed design from the other reported MS antennas.

Author action: The difference between the proposed antenna and other MS antennas is further emphasized in Abstract of the revised manuscript.

Concern # 2: Please show MS surface current distribution of mode 3&4.

Author response: According to the Reviewer’s comments, the surface current distribution and radiation pattern of Mode-3 and -4 are presented in Fig. 1R. As seen, the currents of these modes flow in different directions, resulting in a null in the broadside direction. Meanwhile, this paper focuses on designing an antenna with broadside radiation. Therefore, Mode-3 and -4 are not useful for the paper target.

Fig. 1R. Current and radiation of Mode-3 and Mode-4.

Author action: A brief discussion about Mode-3 and -4 is added to Paragraph 2, Subsection “MS layer design”, Section “Narrow-band MS antenna” of the revised manuscript.

Concern # 3: Why the surface current distribution in Fig.2 for MS1&2 is orthogonal?

Author response: For the arbitrary shape of metallic object, the current flowing on this object can come from any direction, leading to different radiation characteristics (as shown in Fig. 1 and Fig. 1R). Depending on the excited source, the proper mode on the object will be excited.

In this paper, the MS layer has a square shape, and the current can flow in horizontal, vertical, or diagonal direction, and so on. Mode-1 and Mode-2 are the fundamental modes, whose currents flow along x- and y-direction. It is similar to the Conventional Rectangular Microstrip Patch antenna. Thus, these modes are orthogonal. Due to equal size in both directions, the resonance frequencies or the modal significances of these modes are identical.

Noted that this behavior is very common for MS layer. It can be found in many other published papers [1R–4R].

[1R] https://doi.org/10.1109/LAWP.2019.2917758.

[2R] https://doi.org/10.1109/TAP.2018.2860121.

[3R] https://doi.org/10.1109/APWC.2019.8870430.

[4R] https://doi.org/10.1038/s41598-024-58794-1.

Concern # 4: How authors are able to determine the feeding point of antenna in Fig. 3.

Author response: In Fig. 3, the MS layer acting as the primary radiating aperture of the whole antenna is coupled with the rectangular patch acting as the primary radiating source of the antenna. The feeding position will be along this rectangular patch. Based on simulation, a proper feeding position can be found.

Author action: A brief discussion about the feeding point is added to Paragraph 1, Subsection “Antenna design and performance”, Section “Narrow-band MS antenna” of the revised manuscript.

Concern # 5: Authors must show the MS graphs for different modes in case of Fig. 3 and Fig.8 antennas and please comment on them.

Author response: The authors would like to thank the Reviewer for the very constructive comment. The CMA on the structures in Fig. 3 and Fig. 8 are implemented. Note that there are many modes with different radiation patterns within the investigated frequency range, only the dominant modes with broadside direction are shown to avoid confusion.

- Firstly, the CMA on the antenna in Fig. 3 is considered, as shown in Fig. 2R(a). As seen, the resonant of Mode-1 is around 5.3 GHz, which is quite close to the simulated data in Fig. 2R(b) (antenna with coaxial cable excitation). The currents of Mode-1 and Mode-2 flow along x- and y-direction. Due to the asymmetrical geometry, the resonances of Mode-1 and Mode-2 are not identical.

- Secondly, the CMA on the antenna in Fig. 8 is considered, as shown in Fig. 3R. As seen, two resonances around 4.9 and 5.6 GHz are produced. The resonances of Mode-1 and -2 (along x- and y-direction) are slightly difference due to the un-equal length of the crossed patch. Meanwhile, Mode-7 and -8 have currents flowing along the diagonal direction. Thus, the resonances or modal significances of these modes are almost identical. These resonances are consistent with the simulated data in Figs. 9 and 10 (in the revised manuscript), in which the lower resonance (5.1 GHz) has current flowing along the vertical direction and the higher resonance (5.3 GHz) has current flowing along the diagonal direction.

(a)

(b)

Fig. 2R. (a) CMA on the MS with single rectangular patch and (b) simulated antenna with coaxial cable excitation.

(a)

(b)

(c)

Fig. 3R. CMA on the MS with crossed patch. (a) modal significance, (b) surface current and radiation pattern. And (c) simulated antenna with coaxial cable excitation.

Author action: The CMA on the narrow-band and wide-band MS antennas are added to the revised manuscript as Fig. 5 and Fig. 11.

Reviewer 2:

Concern # 1: The authors must describe the significant resonant modes for the proposed design.

Author response: The authors would like to thank the Reviewer for the constructive comment. The modal significance based on the CMA of the 2 × 2-unit cell MS is shown in Fig. 4R(a). Here, four possible operating modes are observed in the frequency range from 4.8 to 5.6 GHz. Among them, only Mode-1 and Mode-2 have broadside direction patterns (as demonstrated in Fig. 2 in the revised manuscript). Therefore, these modes are utilized in the proposed antenna (Fig. 3 and Fig. 8). The other modes, Mode-3 and -4, have different radiation patterns, shown in Fig. 4R(b).

(a)

(b)

Fig. 4R. (a) Modal significance and (b) surface current and 3-D radiation pattern of the 2 × 2-unit cell MS.

Author action: The surface current and 3-D radiation pattern of Mode-3 and -4 are included in Fig. 2 of the revised manuscript. A brief discussion on these modes is also added to Paragraph 2, Subsection “MS layer design”, Section “Narrow-band MS antenna” of the revised manuscript. Additionally, the CMA on the MS antenna with single rectangular patch and crossed patch is also added to Sections “Narrow-band MS antenna” and “Wideband MS antenna”.

Concern # 2: Some parametric investigations on design parameters must be shown and discussed.

Author response: In this paper, the design steps can be summarized as shown in Fig. 5R. For Design-1, the key parameters related to lp, g, W have been shown and discussed in Subsection “Antenna operation characteristics” of the Section “Narrow-band MS antenna”. For Design-2, the effect of the horizontal patch on the bandwidth enhancement is also discussed in Section “Wideband MS antenna with SIW cavity”. For the final Design-3, the SIW cavity requires multiple vias in proximity to form an electric wall, which is better with a greater number of vias. Meanwhile, the distance between the vias depends on the radius of the vias. Therefore, the authors believe that all necessary key design parameters have been included in the manuscript.

Fig. 5R. Design steps to achieve final realization of the proposed antenna.

Author action: For better understanding, the discussion about the design of SIW cavity is added to Paragraph 1, Section “Wideband MS antenna with SIW cavity”.

Concern # 3: How the coupling between metacells exists, an explanation is desired.

Author response: The authors do NOT fully understand the Reviewer question, if possible, the authors will response properly in the next revision stage.

Here, the unit cells are positioned in proximity. When the cell is excited, it will capacitively coupled with its adjacent unit cells.

Concern # 4: What is the utility for asserting SIW technique?

Author response: Basically, the small antenna suffers from high back radiation due to the propagation of the surface wave in the substrate and then diffract at the edges of the ground plane. Therefore, SIW technique is used to reduce back radiation. The effectiveness of the SIW in the proposed design is demonstrated in Fig. 13b of the revised manuscript.

Author action: The utility of the SIW cavity is emphasized in Paragraph 3, Section “Wideband MS antenna with SIW cavity” of the revised manuscript.

Concern # 5: A comparative study analysis with other literatures must be shown for radiation efficiency and number of antenna layers.

Author response: The no. of antenna layers has been included in the comparison table as “No. of layers” column. The radiation efficiency values are not mentioned in the other papers. Thus, the authors do not include this parameter.

Author action: For Reviewer’s convenience, the number antenna layers is highlighted in the comparison table. Additionally, the radiation efficiency of the proposed antenna is discussed in Paragraph 2, Section “Experimental results and discussion” of the revised manuscript.

---

## [Decision Letter · Decision Letter 1]

10 Mar 2025

High front-to-back ratio metasurface antenna with single-layer and compact size for WLAN applications

PONE-D-25-03828R1

Dear Dr. Tran-Huy,

We’re pleased to inform you that your manuscript has been judged scientifically suitable for publication and will be formally accepted for publication once it meets all outstanding technical requirements.

Kind regards,

Sachin Kumar, Ph.D.

Academic Editor

PLOS ONE

Additional Editor Comments (optional):

The authors have carefully addressed the reviewers' comments, and the manuscript is accepted for publication.

Reviewers' comments:

Reviewer's Responses to Questions

**Comments to the Author**

1. If the authors have adequately addressed your comments raised in a previous round of review and you feel that this manuscript is now acceptable for publication, you may indicate that here to bypass the “Comments to the Author” section, enter your conflict of interest statement in the “Confidential to Editor” section, and submit your "Accept" recommendation.

Reviewer #1: All comments have been addressed

Reviewer #2: All comments have been addressed

2. Is the manuscript technically sound, and do the data support the conclusions?

Reviewer #1: Yes

Reviewer #2: Yes

3. Has the statistical analysis been performed appropriately and rigorously? 

Reviewer #1: N/A

Reviewer #2: Yes

4. Have the authors made all data underlying the findings in their manuscript fully available?

Reviewer #1: Yes

Reviewer #2: Yes

5. Is the manuscript presented in an intelligible fashion and written in standard English?

Reviewer #1: Yes

Reviewer #2: Yes

6. Review Comments to the Author

Reviewer #1: Authors have addressed all my concerns in the paper. Manuscript can be accepted in the present form.

Reviewer #2: The authors have successfully addressed the queries. I have no further comments. The manuscript can be accepted for publication. The revisions has all the updates and the authors have successfully addressed.

7. PLOS authors have the option to publish the peer review history of their article (what does this mean?). If published, this will include your full peer review and any attached files.

Reviewer #1: No

Reviewer #2: **Yes: **Asutosh Mohanty

---

## [Editor Report · Acceptance letter]

PONE-D-25-03828R1

PLOS ONE

Dear Dr. Tran-Huy,

I'm pleased to inform you that your manuscript has been deemed suitable for publication in PLOS ONE. Congratulations! Your manuscript is now being handed over to our production team.

Kind regards,

on behalf of

Dr. Sachin Kumar

Academic Editor

PLOS ONE